# Germline polymorphisms in the immunoglobulin kappa and lambda loci underpinning antibody light chain repertoire variability

Eric Engelbrecht [1], Oscar L. Rodriguez[1,2], William Lees [3], Zach Vanwinkle[1], Kaitlyn Shields[1], Steven Schultze[1], William S. Gibson[1], David R. Smith[1], Uddalok Jana[1], Swati Saha [1], Ayelet Peres [4], Gur Yaari[4], Melissa L. Smith[1] ✉ & Corey T. Watson [1] ✉

Variation in antibody (Ab) responses contributes to variable disease outcomes and therapeutic responsiveness, the determinants of which are incompletely understood. This study demonstrates that polymorphisms in immunoglobulin (IG) light chain loci dictate the composition of the Ab repertoire, establishing fundamental baseline differences that influence functional Ab-mediated responses. Using long-read genomic sequencing of the IG kappa (IGK) and IG lambda (IGL) loci, we resolve genetic variation, including structural variants, single nucleotide variants, and gene alleles. By integrating these genetic data with Ab repertoire profiling, we find that all forms of IG germline variation contribute to inter-individual gene usage differences for >70% of light chain genes in the repertoire, directly impacting the amino acids of expressed light chain transcripts. The genomic locations of usage-associated variants in both intergenic and coding regions indicate that IG polymorphisms modulate gene usage via diverse mechanisms, likely including the modulation of V(D)J recombination, heavy and light chain pairing biases, and transcription/translation. Finally, relative to IGL, IGK is characterized by more extensive linkage disequilibrium and genetic co-regulation of gene usage. These results firmly establish the critical contribution of IG light chain polymorphism in Ab repertoire diversity, with important implications for investigating Ab responses in health and disease.

Antibodies (Abs) are critical components of the adaptive immune system and are one of the most diverse protein families in the human body. The circulating Ab repertoire comprises hundreds of millions of unique antibodies[1,2], and its composition varies significantly between individuals[1–3]. The variability likely contributes to the diverse Ab responses observed across clinical settings, including infection[4–8], autoimmunity[9], and cancer[10]. Identifying the factors that contribute to variation in B cell-mediated immunity will inform disease diagnosis and treatment.

Human Abs are composed of two pairs of identical 'heavy' chains and 'light' kappa or lambda chains, encoded by genes located at three

[1]Department of Biochemistry and Molecular Genetics, University of Louisville School of Medicine, Louisville, KY, USA. [2]Department of Microbiology, Icahn School of Medicine at Mount Sinai, New York, NY, USA. [3]Clareo Biosciences, Louisville, Kentucky, USA. [4]Department of Pathology, Yale School of Medicine, New Haven, CT, USA. ✉e-mail: ml.smith@louisville.edu; corey.watson@louisville.edu

primary loci in the genome: the immunoglobulin (IG) heavy chain locus (IGH; 14q32.33), and the IG lambda (IGL; 22q11.2) and kappa (IGK; 2p11.2) loci[11]. Across the IG loci, there are >240 phylogenetically related functional/open reading frame variable (V), diversity (D) (specific to IGH), and joining (J) genes[12–15]. Selection of individual IG heavy and light chain genes during V(D)J recombination is a foundational step in the process of Ab generation. Increasing evidence indicates that genetic variation within the IG loci modulates the generation of the formation of the human Ab repertoire, contributing to the observed receptor diversity seen between individuals. This was initially supported in twin studies, which demonstrated that both naïve and antigen-stimulated Ab repertoires possess heritable characteristics[16–18]. Additionally, non-coding and coding IG heavy chain[12,19–26] and light chain[20,27–29] germline variants have been shown to affect Ab gene usage and antigen specificity.

Utilizing matched adaptive immune receptor repertoire (AIRR)-seq and comprehensive long-read sequencing-based genotyping of IGH in a cohort of 154 individuals, we recently demonstrated that approximately half of common germline variants in IGH were associated with variation in usage frequencies of the majority of IGHV, IGHD, and IGHJ genes within the IgM (naïve-enriched) repertoire[12]. Subsequently we showed that these genetic variants contribute to repertoire variation in early B cell developmental stages in the bone marrow, indicating direct impacts on V(D)J recombination[30]. This has raised the prospect that variants within the IGK and IGL loci exert similar effects on the formation of the light chain repertoire, and will associate with inter-individual Ab variation in the periphery. The heavy and light chains of an Ab must be paired and compatible to achieve specificity and functionality, and both heavy and light chains contribute to antigen binding. The identification and characterization of antigen-specific and disease-associated Abs requires comprehensive models of naïve and antigen-experienced repertoires, for which a detailed understanding of genetic variation in all three loci is essential. To this end, it is critical to recognize that IG loci are enriched with structural variants (SVs), including segmental duplications and insertions, limiting the utility of short-read sequencing to characterize genetic variation[12–15,31–34]. We previously demonstrated that long-read sequencing of diverse IGH[12,13,33], IGL[14], and IGK[15] haplotypes identifies genetic variation not recorded in reference databases, including catalogues of single-nucleotide variants (SNVs) and gene alleles.

Here, we pair long-read genomic sequencing of IGK and IGL with AIRR-seq at population-scale to identify cis-acting variants that explain inter-individual variation in light chain Ab repertoire features. We find that genetic variants in IGK and IGL associate with gene usage frequency for the majority of light chain V and J genes. These associations between germline polymorphism and gene usage persisted even in antigen-experienced Ab repertoires. Analysis of lead variants revealed mechanisms by which genetic sequence can impact gene usage frequencies, including missense and nonsense substitutions, as well as substitutions in regulatory elements, such as recombination signal sequences. We find distinct structures of linkage disequilibrium (LD) in IGK and IGL, with relatively high LD in IGK associating with coordinated usage of multi-gene clusters. Finally, we demonstrate that genetic effects on gene usage contribute to amino acid variation in V genes, as well as physicochemical properties of the CDR3, linking germline variants to Ab features that contribute to antigen binding.

## Results
### Long-read genomic sequencing and genotyping of IGK and IGL loci and expressed light chain antibody repertoire sequencing
We combined targeted long-read sequencing of IGK and IGL loci in 177 healthy individuals with newly and previously[12] generated AIRR-seq for nearly all donors in the cohort (IGK, n = 164, IGL, n = 168). Donors ranged in age from 18 to 57 years (mean: 32.4), representing both biological sexes (male, n = 87; female, n = 84), and diverse genetic

ancestry groups (Supplementary Data 1). Using our previously published method[13], we performed targeted long-read single molecule real-time (SMRT) sequencing of the IGK proximal and distal regions[15], and the IGL locus[14] (see Supplementary Material). From these data, we generated sample-level SNV and SV callsets, and IGK/L gene and allele germline sets (see Supplementary Material). Importantly, this dataset allowed for the identification of uncatalogued variants, including >300 germline IG alleles as well as SNVs and SVs (Supplementary Fig. 2).

To profile expressed IGK and IGL transcripts, AIRR-seq data was generated using 5' rapid amplification of complementary DNA ends (5' RACE) on total RNA isolated from PBMCs. With germline IGKV, IGKJ, IGLV, and IGLJ alleles for each individual, we limited V and J germline allele calls to those present in the germline on a per-individual basis. To enrich for antigen-naïve BCR sequences, we selected those containing V and J segments that matched germline allele sequences with 100% identity, unlikely to have undergone somatic hypermutation (SHM) (i.e. unmutated). The opposite approach was used to enrich for antigen-experienced BCR sequences, for which either the J or V (or both) segment varied from the germline allele sequence (i.e., were mutated). Importantly, personalized germline sets allowed us to account for the presence of previously undocumented alleles, and thus more accurately infer SHM. The usage frequencies of V and J genes among all unmutated or mutated unique BCR sequences were calculated for each individual. Together, these datasets allowed us to resolve comprehensive variant callsets to perform genetic association analyses with gene usage variation observed in the expressed light chain Ab repertoires.

### Light chain gene usage is strongly associated with common IGK and IGL genetic variants in both antigen naïve and experienced repertoires
Throughout the genome, genetic variation has been associated with molecular traits such as gene expression and splicing[35–38]. We previously demonstrated that genetic variants in the IGH locus mediate the composition of peripheral IgM and IgG repertoires through effects on IGHV, IGHD, and IGHJ gene usage[12]. Here, we followed this same quantitative trait locus (QTL) framework to test if light gene usage was associated with IGK and IGL variant genotypes in cis. Allele assignments to AIRR-seq reads were derived from a personalized germline allele set for each individual. This permitted disambiguation of IGK gene paralogs for individuals wherein each allele of a proximal paralog was distinct from each allele of the distal paralog, including IGKV1-12 and IGKV1D-12, IGKV1-13 and IGKV1D-13, and IGKV6-21 and IGKV6D-21 (Supplementary Data 5). Paralog pairs for which at least 160 individuals could not be disambiguated included IGKV1-33/1D-33, IGKV1-37/1D-37, IGKV1-39/1D-39, IGKV2-28/2D-28, and IGKV2-40/2D-40, and are referred to as ambiguous or "ambi" (e.g. IGKV1-39/1D-39 is IGKV1-39ambi).

We performed genetic association tests on unmutated ("antigen naïve") and mutated ("antigen experienced") sets separately to identify cis effects in each of the two repertoire sets. In the unmutated IGK repertoire, after Bonferroni multiple-testing correction (P < 5.0e−05), we identified 2352 unique variants (2350 SNVs, 2 SVs) that were statistically associated with gene usage changes in 21 IGKV and 3 IGKJ genes (Fig. 1, Supplementary Data 6). In the unmutated IGL repertoire, a set of 911 unique variants (910 SNVs, 1 SV) were associated with gene usage changes in 22 IGLV and 3 IGLJ genes (Fig. 1, Supplementary Data 6). Notably, a large fraction of the genes identified in both the IGK (n = 13 genes) and IGL (n = 19 genes) unmutated repertoires also had significant gene usage QTLs (guQTLs) in the mutated repertoires (Supplementary Figs. 7–9, Supplementary Data 7). However, guQTLs in the unmutated repertoires tended to have lower P values and explain more variance (R²) in gene usage (Supplementary Fig. 7). We also noted stronger genetic effects in IGK relative to IGL; this included the observation that overall genetic similarity among subjects associated with more highly correlated IGK gene usage, a signal that was blunted

 

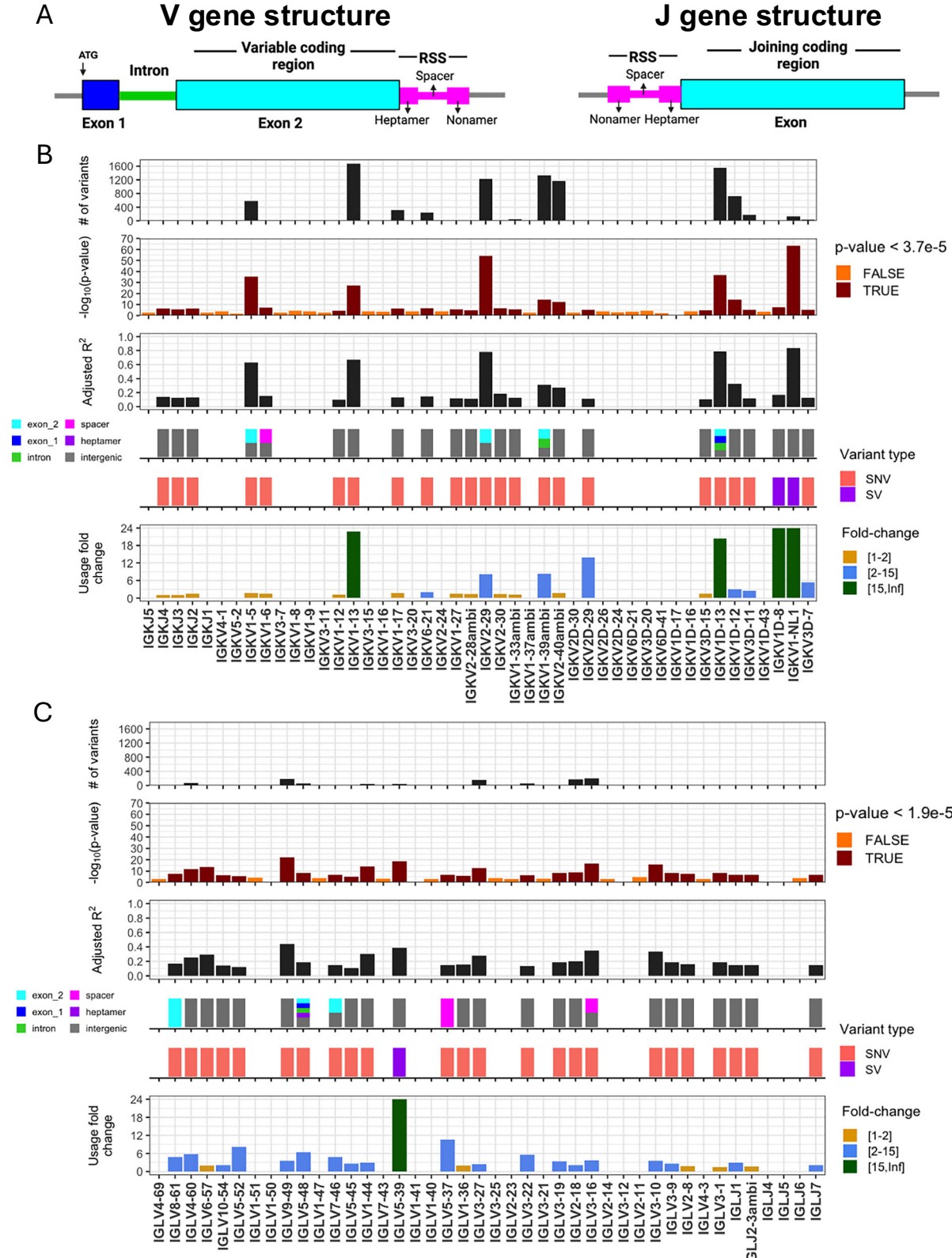

**Fig. 1 | IGK and IGL variants impact gene usage in the naïve Ab repertoire.**
**A** General structure of V and J genes in the IGK and IGL loci, including location of the recombination signal sequences (RSS). **B, C** Per gene (x axis, all panels) statistics from linear regression guQTL analysis for the repertoire of unmutated IGK (**B**) and IGL (**C**) light chains, including: (i) the number of associated variants after Bonferroni correction (IGK; $P < 3.7e{-}5$, IGL; $P < 1.9e{-}5$), (ii) $-\log_{10}(P$ value) of the lead guQTL, (iii) adjusted $R^2$ for variance in gene usage explained by the lead guQTL, (iv) the location and (v) type of variant for the lead guQTL and (vi) the fold change in mean gene usage between genotype groups at the lead guQTL. Summary statistics are provided in Supplementary Data 6. (**A**) Created using (https://BioRender.com/l3h23u7).

in IGL (see Supplemental Material, Supplementary Fig. 10). Together, these results show that usage of IG light chain genes is broadly impacted by germline genetic variants, and while these genetic effects are more prominent in the antigen naïve repertoire, for many genes, those effects persist even following antigen exposure.

## Genomic locations of guQTLs implicate genetic roles in coding and non-coding driven processes underlying antibody repertoire formation

Features of IG J genes include a single exon and RSS sequence, whereas V genes include an RSS sequence, first exon, intron, and second exon that encodes the antigen binding variable region (V-region) (Fig. 1A). While the majority of lead guQTLs across the unmutated repertoires were intergenic (n = 19, IGK; n = 20, IGL), coding lead variants were identified for 7 IGK genes (IGKJ3, IGKV2-29, IGKV1-5, IGKV1-13, IGKV1D-13, and IGKV1-39ambi) (Fig. 1B, Supplementary Fig. 8) and 4 IGL genes (IGLV8-61, IGLV5-48, IGLV7-46, IGLV3-21) (Fig. 1C, Supplementary Fig. 8). In addition, 1 IGK lead guQTL and 5 IGL guQTLs fell within RSSs; these were lead variants for IGKV1-6, IGLV5-37, IGLV3-16, IGLV1-44, IGLV3-19, and IGLV5-48 (Fig. 1B, C, Supplementary Fig. 8).

Examples of lead guQTLs in coding, RSS, and intergenic regions are shown in Fig. 2, including IGKV2-29, IGKV1-5, IGLV3-16, and IGLV9-49. The SNV-driven guQTL in this dataset with the lowest P value was for IGKV2-29 (P value = 7.4e−55, Fig. 2A). This variant introduced a stop codon in V-region amino acid position 93 (Fig. 2B), resulting in decreased usage of IGKV2-29 (Fig. 2C). We also identified a lead guQTL associated with missense variants. In the case of IGKV1-5, two linked lead guQTLs ($r^2 = 1$) within codon 50 associated with a lysine to aspartic acid (AAG → GAT, K50D) change, resulting in an alteration of residue charge (Fig. 2D, E). Individuals homozygous for K50 alleles, which represented six different IGKV1-5 coding alleles in this cohort, had lower gene usage (Fig. 2F, G). As an example of a guQTL in the RSS, two lead variants in perfect LD were identified at positions 8 and 23 of the spacer for IGLV3-16 (Fig. 2H–J). The reference haplotype had a C at position 8, which was represented among consensus bases (C and T) at this position (Supplementary Fig. 11), whereas the alternate haplotype had a G (Fig. 2I). Among the C/C and G/G lead guQTL genotype groups, IGLV3-16 usage varied 3.7-fold on average (Fig. 2J). As noted above, the majority of lead guQTLs in this dataset were in non-coding regions. For example, the lead guQTL for IGLV9-49 was 86 bp upstream of the first exon, and guQTLs were not identified in coding sequence or the RSS (Fig. 2K). Mean IGLV9-49 usage varied by 3.5-fold between homozygous-reference and homozygous-alternate individuals at this lead guQTL (Fig. 2L).

Consistent with previous observations in IGH[12], many guQTLs within IGL overlapped curated transcription factor binding sites, representing an enrichment over background SNVs (see Supplementary Material, Supplementary Figs. 12, 13), suggesting likely roles for non-coding variants in the regulation of V(D)J recombination. Additionally, in IGK, we noted that both coding and non-coding regulatory variants altered proximal and distal gene usage biases (see Supplementary Material, Supplementary Figs. 14, 15).

Finally, in addition to SNV guQTLs, SVs resulting in gene copy number changes also made significant impacts on gene usage. Specifically, SVs were lead guQTLs for the genes IGKV1-NL1, IGKV1D-8, and IGLV5-39. In all cases, differential usage between genotypes followed an additive model in which gene usage increased with every additional haploid gene copy (Fig. 2M). The lead variant associated with IGKV1D-8 usage was the SV deletion of the entire IGKV distal region (see Supplementary Fig. 4). We noted that the number of diploid IGLJ2-3 cassette copies associated with the usage of IGLJ1 and IGLJ2-3ambi (Supplementary Fig. 16); however, this CNV was not the lead QTL for these genes. The complexity of this SV will likely require analysis in larger cohorts and more detailed assessment of potential haplotype-specific effects.

In summary, these results indicate that many forms of genetic variation are associated with gene usage variation in the IGK and IGL repertoire. The variable localization of guQTLs in intergenic, RSS, and coding regions implicates causative roles for these genetic variants in plausibly regulating V(D)J recombination, transcription, and translation, as well as contributing to differential heavy-light chain pairing dynamics and antigen selection.

## guQTLs within large linkage disequilibrium blocks in IGK create expansive networks of genes with correlated usage

In our previous study of IGH guQTLs, we observed that many SNVs were associated with the usage of individual genes. This included instances in which genes and associated guQTLs extended 10's to 100's of Kb; notably, these genes exhibited correlated usage patterns[12], suggestive of coordinated gene regulation. We sought to investigate whether similar features were present in the IGK and IGL loci.

First, within the unmutated repertoire, we observed a greater number of guQTLs in IGK compared to IGL (Fig. 1B, C). This was not simply explained by the number of SNVs genotyped in the two loci, as we identified twice as many common variants in IGL relative to IGK; among all common SNVs in each locus, 84.2% in IGK and 17.5% in IGL were significantly associated with usage of at least one gene (Fig. 3A). Compared to IGL, we found that a larger fraction of IGK guQTL variants were shared between at least two genes (n = 1995, 83.3%) (Fig. 3B). Likewise, at gene-level, a greater number of IGK guQTL genes shared at least one significant variant with >5 other genes (Fig. 3C).

To visualize these relationships between genes and guQTLs, we constructed networks in which nodes represented genes and edges represented connections between genes sharing at least one guQTL SNV. From these networks, we identified multi-member cliques, in which 2 or more genes were connected by at least one shared guQTL. For IGK, 21 of the 24 guQTL genes formed a single super clique, with embedded subcliques in which all genes were connected to one another through guQTL variants (Fig. 3D). Demonstrative of inter-connected gene usage, the largest subclique was composed of 9 guQTL genes associated with a single guQTL SNV (Fig. 3E). In contrast to IGK, 13 of the 25 guQTL IGL genes were represented by 4 distinct cliques, all of which were smaller than the large clique observed in IGK and disconnected from one another (gene membership range = 2–5; Fig. 3D).

The stark difference in IGK and IGL clique sizes (Fig. 3D) suggested likely differences in the genetic haplotype structure between the two loci. To explore this, we estimated pairwise linkage disequilibrium (LD) between all common SNVs (MAF ≥ 5%) and determined blocks of LD[39,40] (Supplementary Data 9, see Methods). LD was more extensive in IGK (Fig. 4A) relative to IGL (Fig. 4B), with LD blocks >20 Kbp comprising 53.5% and 12.9% of the IGK and IGL loci, respectively, (Fig. 4C, Supplementary Data 10). The three largest LD blocks in IGK were 122 Kbp, 110 Kbp, and 76 Kbp, compared to the three largest LD blocks in IGL that were 34 Kbp, 26 Kbp and 24 Kbp (Fig. 4D). As expected, the number of SNVs per block was positively correlated in both loci. The overall density of common SNVs was about 1.8 times higher in IGL relative to IGK (Fig. 4E). Additionally, larger sets of genes in IGK fell within large LD blocks (Fig. 4F; Supplementary Fig. 17, 18). IGK guQTL SNVs were also more frequently in large LD blocks (Fig. 4G).

These data demonstrate that a larger proportion of IGK sequence, genes, and guQTLs are contained within large LD blocks as compared to IGL. The overlap of LD blocks with guQTL and gene cliques suggests that extended haplotype structures within both loci likely contribute to coordinated gene regulation.

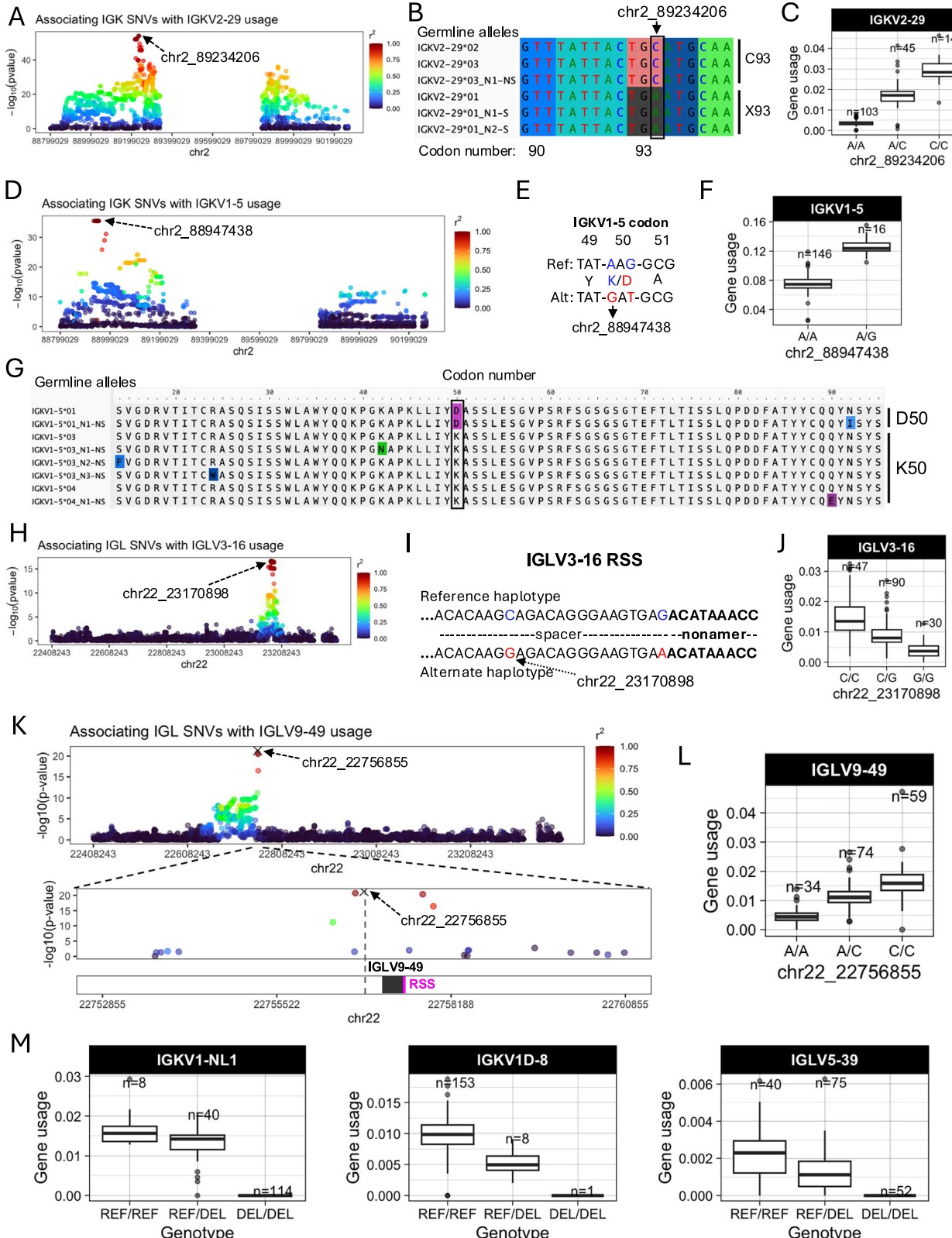

### guQTLs are linked to missense variation in coding regions and physicochemical CDR3 properties in the IGK and IGL repertoires

Together, the data presented so far demonstrate that genetic variants within IGK and IGL associate with shifts in gene usage in the light chain repertoire. While the majority of lead guQTLs in both loci occurred in intergenic space, we wanted to see whether genetically driven usage shifts also associated with (1) germline changes in V

gene coding sequence spanning complementarity determining and framework regions (CDR1, CDR2, FWR1, FWR2, and FWR3); and (2) amino acid properties of CDR3 sequences spanning germline codons and junctions of recombined V and J genes. We reasoned that such associations would link changes in gene usage to BCR features likely to impact preferential pairing of available heavy and light chains and antigen binding. This has direct relevance to

**Fig. 2 | Examples of coding and non-coding lead guQTLs. A** Manhattan plot showing the −log10(P value) for all SNVs in IGK tested for association with usage of *IGKV2-29*, with SNVs colored according to LD ($r^2$). **B** Sequence alignment of the germline *IGKV2-29* alleles in this cohort from codons 90 to 95, with the lead variant indicated. Alleles encoding C93 and X93 (STOP codon) alleles are indicated. **C** Boxplot of *IGKV2-29* usage in lead guQTL genotype groups. **D** Manhattan plot of associations (−log10(P value)) between all IGK SNVs and usage of *IGKV1-5*, with SNVs colored according to LD ($r^2$) with the lead variant. **E** Sequence alignment of the reference and alternate haplotypes at the lead guQTL, with two missense variants in perfect LD in codon 50 indicated, resulting in K50D in the alternate haplotype. **F** Boxplot of *IGKV1-5* usage in lead guQTL (shown in (**D**, **E**)) genotype groups. **G** Alignment of translated germline *IGKV1-5* alleles with codon 50 boxed. **H** Manhattan plot of associations (−log10(*P* value)) between all IGL SNVs and usage

of *IGLV3-16*, with SNVs colored according to LD ($r^2$) with the lead variant. Two lead variants in perfect LD are in the RSS spacer. **I** Sequence of the RSS spacer in reference and alternate lead guQTL haplotypes. **J** Boxplot of *IGLV3-16* usage in lead guQTL genotype groups. **K** (Top) Manhattan plot of associations (−log10(P value)) between all IGL SNVs and usage of *IGLV9-49*, with SNVs colored according to LD ($r^2$) with the lead variant (marked with an X). (Bottom) Zoom-in on an 8 Kbp window centered on *IGLV9-49* with the lead non-coding variant indicated. **L** Boxplot of *IGLV9-49* usage in lead guQTL genotype groups. **M** Gene usage boxplots of genes for which the lead variant was a deletion ("DEL") SV, including *IGKV1-NL1*, *IGKV1D-8*, and *IGLV5-39*. Boxplots display the median, 25th percentile, 75th percentile, and whiskers that extend up to 1.5 times the inter-quartile range (IQR) from the respective percentiles. Data points outside the whiskers are also plotted.

germline variants contributing to Abs associated with disease and vaccination[20–22,24–26,41].

First, we found that many lead guQTLs were associated with shifts in coding allele usage within the repertoire, representing LD between coding and non-coding SNVs. This was consistent with our previous investigation of IGH guQTLs[12]. Specifically, for 16/26 (62%) tested IGK genes, individuals within different guQTL genotypes exhibited differential coding allele frequencies (two-way Fisher's exact test, Bonferroni; *P* < 0.002). Likewise, in IGL we noted such associations for 8/23 (35%) guQTL genes (two-way Fisher's exact test, Bonferroni; *P* < 0.002) (Fig. 5A, B, Supplementary Data 11). Among these genes, 12/16 (75%) in IGK, and 7/8 (88%) in IGL involved alleles carrying amino acid changes (Fig. 5C, D). For the remaining genes in each locus, genes either exhibited allelic variation, but did not associate with guQTL genotype, or lacked appreciable allelic variation (major allele frequency >95%; Fig. 5C, D). Examples of genes with coding allele variation linked to lead non-coding guQTL variants include *IGKV2-30* and *IGLV10-54*, for which gene alleles were distributed differently among the non-coding lead guQTL genotypes (Fig. 5E, F). In the case of *IGKV2-30*, the *02 allele, which harbored a missense variant in CDR1, was carried by 95.9% of individuals with genotype A/A at the lead guQTL variant, compared to only 7.9% of individuals with genotype G/G (Fig. 5E). Likewise, in the case of *IGLV10-54*, the *04 allele, which harbored an amino acid change in FWR3, was carried by 100% of individuals with guQTL genotype G/C and by 4.5% of individuals with genotype C/C (Fig. 5F).

We next asked whether shifts in gene usage also resulted in shifts in CDR3 physicochemical properties. To do this, we conducted an unbiased test for associations between variation in nine CDR3 properties and genotypes at all variants across IGK and IGL (Supplementary Data 12). In IGK, we identified SNVs associated (Bonferroni, *P* < 3.7e−05) with CDR3 properties of aromaticity, aliphaticity, acidity, polarity, bulk, and GRAVY index (Fig. 6A; Supplementary Fig. 19). Likewise, in IGL, SNVs were associated (Bonferroni, *P* < 1.9e−05) with CDR3 aromaticity, aliphaticity, GRAVY index, bulk, basicity, polarity, charge, and length (Fig. 6B; Supplementary Fig. 19). Lead variants in both loci overlapped guQTLs, linking CDR3 properties with gene usage variation. For example, the lead variant associated with IGK CDR3 aromaticity was also a guQTL for seven IGKV genes (Fig. 6A, C–E) that are part of a previously described network clique (Fig. 3D). Among these genes, we focused on those with usage patterns that were negatively correlated, and thus considered to differentially represent the guQTL genotypes. Specifically, *IGKV1-13/1D-13* usage was highest in individuals with high CDR3 aromaticity. In contrast, the usage of *IGKV2-40ambi*, *IGKV1-39ambi*, and *IGKV1D-12* were highest in individuals with low CDR3 aromaticity (Fig. 6D, E). To determine whether these genes explain CDR3 aromaticity variation between genotype groups, we computed CDR3 aromaticity for BCRs utilizing only *IGKV1-13/1D-13*, *IGKV2-40ambi*, *IGKV1-39ambi*, or *IGKV1D-12* (Fig. 6F). This demonstrated *IGKV1-13/1D-13*-encoded BCRs had higher aromaticity than those encoded by all of the other three genes. This was regardless of

the J gene contribution (Supplementary Fig. 20), indicating the genetic effect on CDR3 aromaticity is through influence on V gene usage.

In IGL, CDR3 aliphaticity and aromaticity shared the same lead variant; this variant was also the lead guQTL for both *IGLJ1* and *IGLJ2-3ambi*, linking genetic regulation of IGLJ gene usage with CDR3 properties (Supplementary Fig. 21). At this variant, A/A individuals had relatively higher CDR3 aliphaticity and IGLJ2-3ambi usage, whereas G/G individuals had relatively higher CDR3 aromaticity and *IGLJ1* usage. Analysis of BCRs using one or the other of these IGLJ genes revealed that sequences containing *IGLJ2-3ambi* have relatively high CDR3 aliphaticity, whereas sequences containing *IGLJ1* have relatively high CDR3 aromaticity (Supplementary Fig. 21). Together, these data link genetic effects on IGLJ gene usage with IGL CDR3 properties.

In summary, IGLV and IGKV genes showed significant variation in coding alleles among lead guQTL genotype groups, indicating LD between non-coding variants and gene alleles. Additionally, we show that variation in gene usage also contributes to biases in CDR3 properties, at least in part explained by contributions of germline encoded amino acids at the 3' of V genes and 5' end of J genes. This is noteworthy, as it indicates that guQTLs not only impact general variation in gene usage, but also have the potential to modulate the availability of germline encoded residues in the baseline unmutated repertoire.

## Discussion

Akin to other hypervariable immune loci, the IG loci exhibit extensive haplotype diversity at the population level, and are among the most structurally complex regions in the human genome[12,14,15,33,42]. This complexity has limited our ability to accurately characterize inter-individual IG haplotype diversity and delineate its role in shaping the composition of the Ab repertoire[41]. Here, by leveraging the strengths of long-read sequencing, we were able to overcome this barrier. We characterized complete genotype callsets for SNVs and SVs across the IGL and IGK loci, and combined these with matched light chain repertoires. First, this comprehensive dataset allowed for the discovery of tremendous population-level genetic variation within the IGK and IGL loci, including descriptions of previously uncatalogued SNVs, SVs, and coding alleles. These discoveries alone will now facilitate significant improvements in existing germline database resources for the IG loci (Peres et al., *in prep*). Second, we were able to directly utilize personalized germline reference sets for each individual to increase the accuracy of V and J gene/allele assignments, including resolution of duplicated paralogs between the proximal and distal duplicated regions of the IGK locus. Third, with these data in hand, we expanded our previous work in IGH[12] to demonstrate that IGK and IGL polymorphisms also contribute to variation in light chain repertoire gene usage and associated changes in the composition and availability of V, J, and CDR3 encoded amino acids among expressed Ab transcripts. Together, these findings solidify the pervasive impact of IG genetics on the adaptive immune system.

Across the IGK and IGL loci, genetic variants were statistically associated with usage variation in the majority of V and J genes. For a

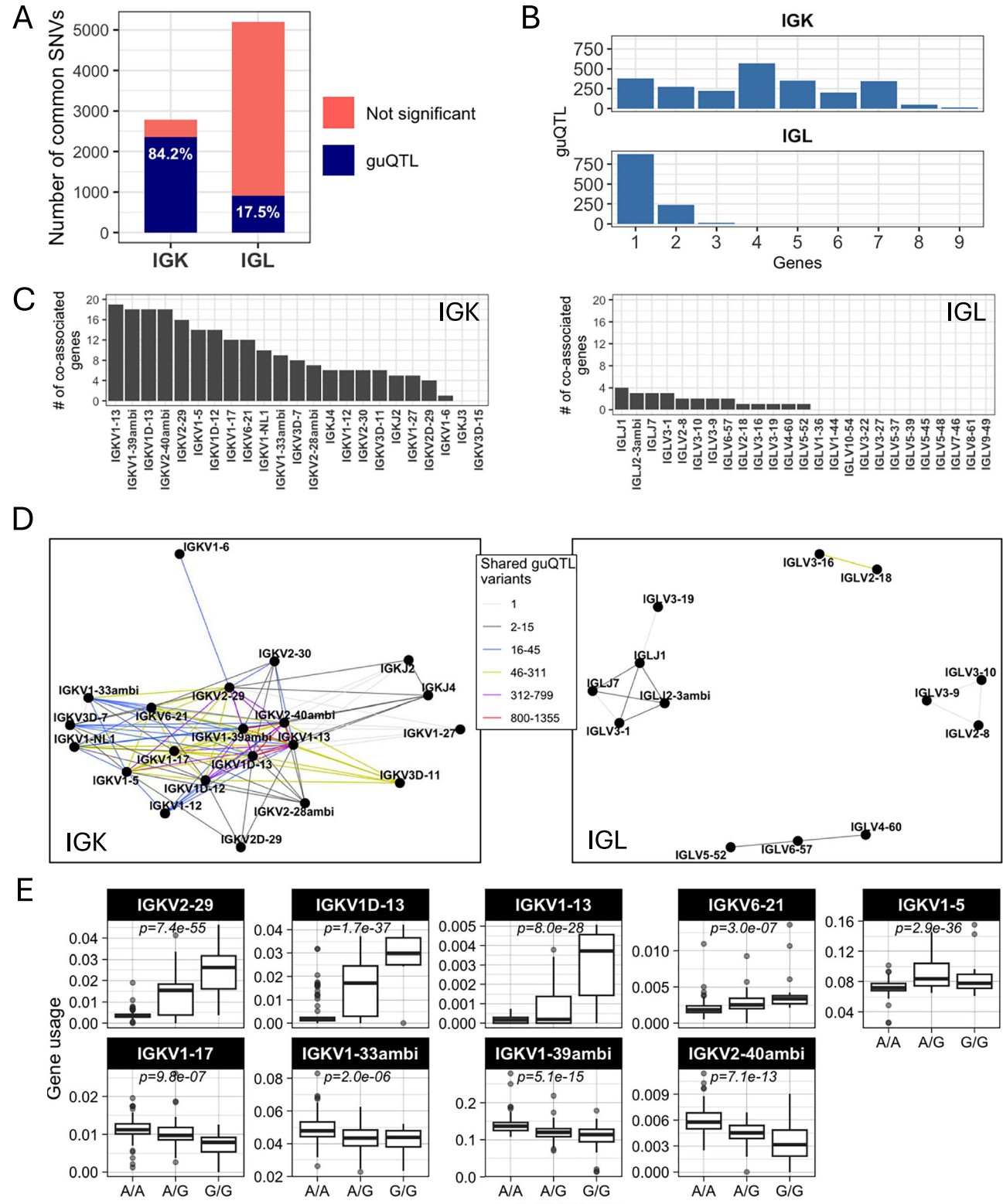

**Fig. 3 | Genetic coordination of IG light chain gene usage is more prevalent in IGK relative to IGL. A** Stacked bar plot showing the proportion of total IGK and IGL common SNVs that are a guQTL. **B** Bar plot showing the number of IGK and IGL SNVs (guQTLs) significantly associated with varying numbers of genes (n = 1–9). For IGK, this includes a large number of SNVs (n = 2049) that were associated with >1 gene. **C** For each gene, the number of genes sharing at least one guQTL variant is plotted for indicated IGK (left) and IGL (right) genes (x-axis). **D** Network analysis identified a large clique of genes and guQTLs in IGK (left) and 4 cliques for IGL

(right), demarcating groups of genes associated with overlapping sets of guQTLs. For each clique, genes are shown as nodes, connected by edges color coded according to the number of shared guQTL variants. **E** Genotype at a single variant associates with usage of the nine plotted IGK genes in the unmutated repertoire (linear regression; *P* value < 3.7e−5). Boxplots display the median, 25th percentile, 75th percentile, and whiskers that extend up to 1.5 times the inter-quartile range (IQR) from the respective percentiles.

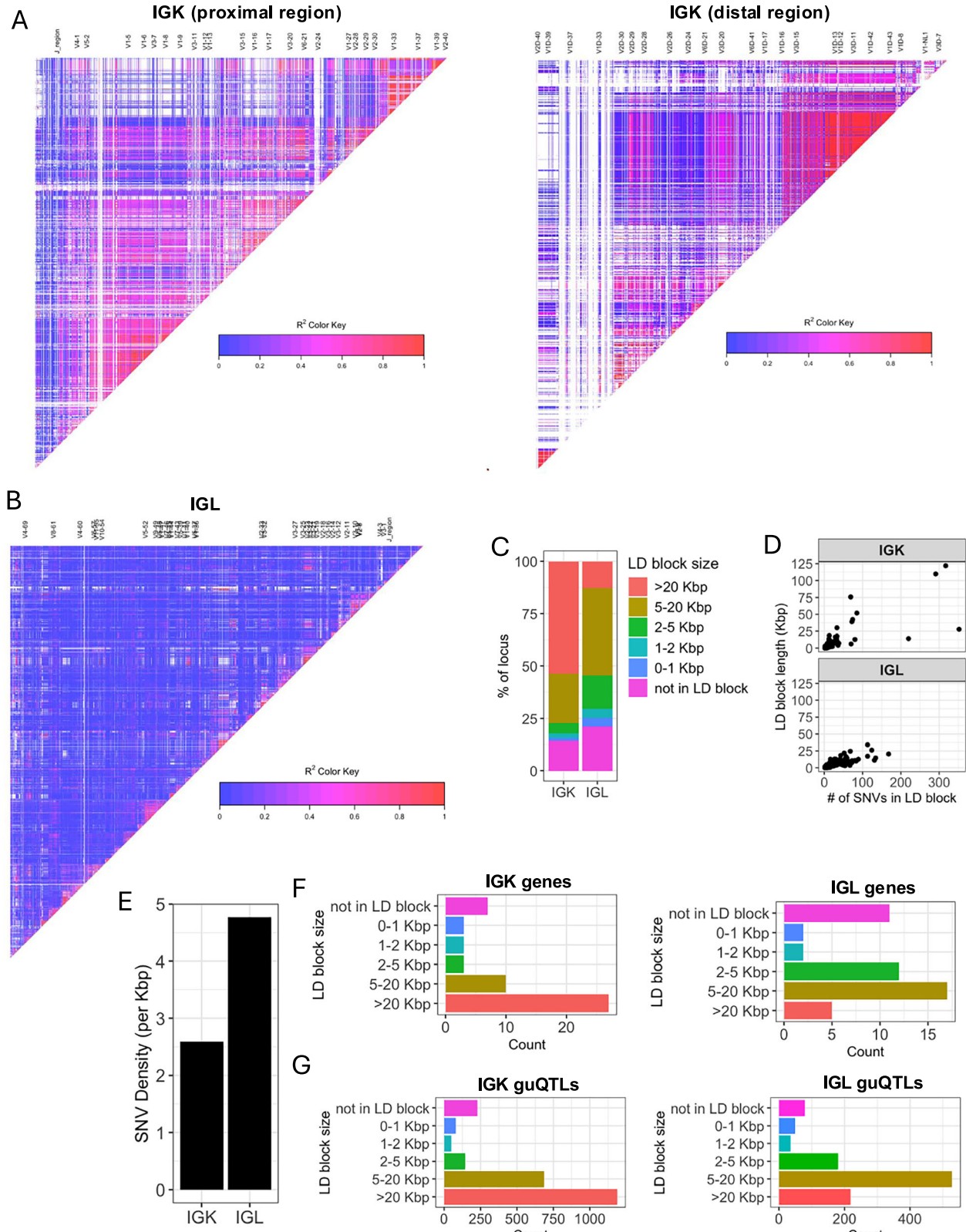

**Fig. 4 | IGK has larger LD blocks and lower density of SNVs relative to IGL.**
**A**, **B** LD heatmaps of the IGK (**A**) and IGL (**B**) loci. LD blocks are illustrated as triangles. **C** Stacked bar plot of the percent of each locus (IGK, IGL) that is within LD blocks of various lengths (colors). **D** Plots of LD blocks in IGK and IGL depicting the length of each block (*y*-axis) and number of SNVs in each blot (*x*-axis). **E** Bar plot of the overall SNV density in the IGK and IGL loci. **F**, **G** Barplots of the counts of IGK or IGL genes (**F**) and guQTL SNVs (**G**) in LD blocks with lengths indicated along *y*-axes.

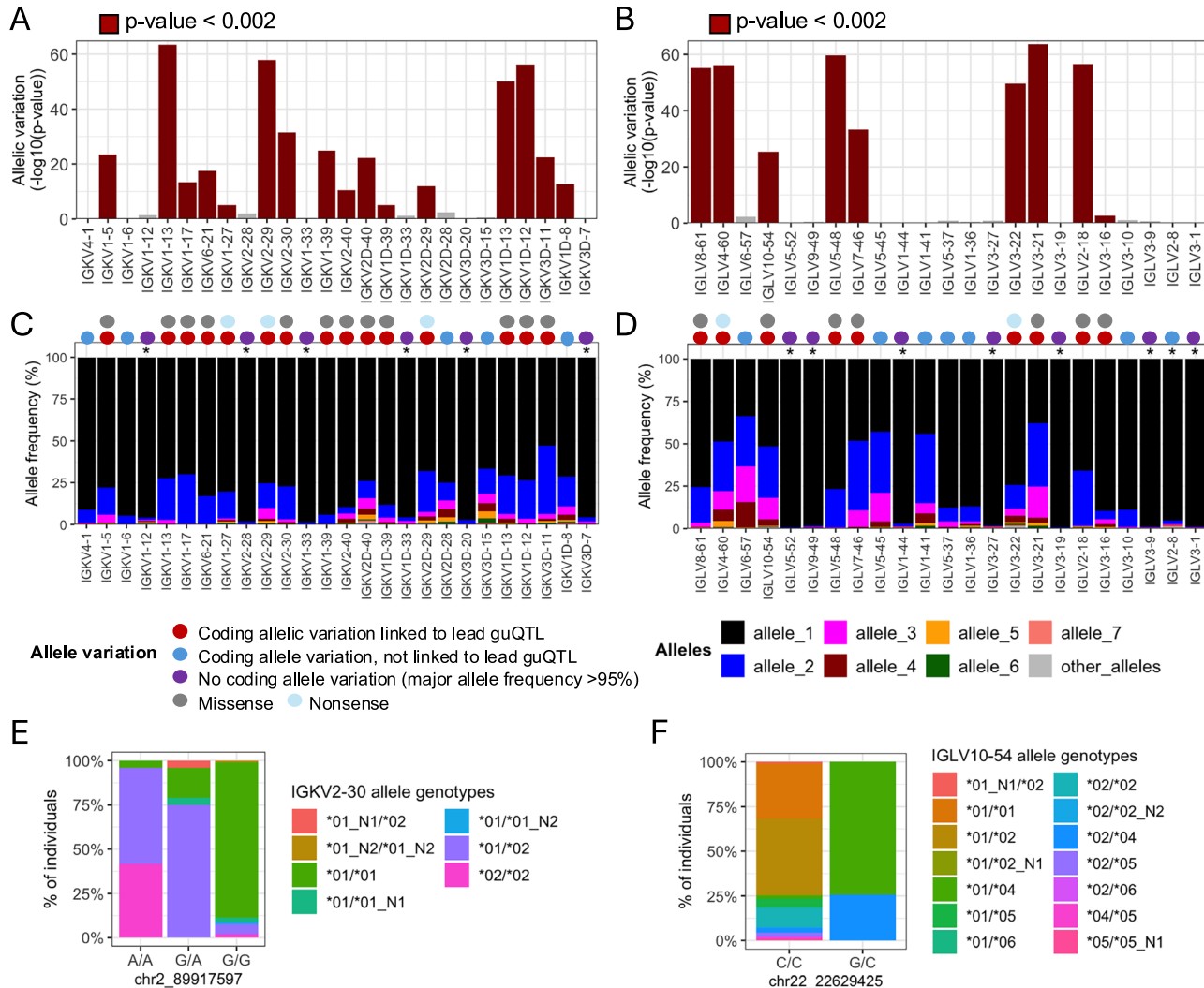

**Fig. 5 | Linkage between IGKV and IGLV coding region alleles and lead guQTL genotypes. A, B** Variation in the proportion of different coding gene alleles among lead guQTL genotype groups was determined by two-way Fisher's exact test for guQTL genes in IGK (**A**) and IGL (**B**). Barplots shows −log₁₀(P value) (Bonferroni; P < 0.002). **C, D** For each gene, the frequency of coding alleles in the cohort is shown, with unique alleles color coded. Genes that lack appreciable allelic variation

(major allele frequency >95%) are indicated with an asterisk. Circles above each gene indicate whether coding allele variation is linked to the lead guQTL. guQTLs linked to coding allele variation are associated with missense or nonsense variants. **E, F** Stacked bar plots showing the distributions of the respective coding allele genotypes across individuals partitioned by guQTL genotype for *IGKV2-30* (**E**) and *IGLV10-54* (**F**).

subset of IGKV genes, for example, we found that even single guQTLs could explain over 75% of the usage variation among donors. Notably, however, although we found that most guQTLs and associated genes were common between unmutated and mutated repertoires, the extent of variance in gene usage explained by guQTLs in the unmutated repertoire was on average higher for both IGK and IGL; V genes also had stronger genetic associations relative to J genes in both loci. The blunted genetic effects in the mutated repertoire may reflect shifts in usage in the memory repertoire driven by interactions with antigen. However, it is notable that even in antigen-experienced repertoires, variation can still be explained by genetic factors, indicating some degree of genetic constraint, consistent with observations in IGH[12,16]. We also noted that on average, $R^2$ values were lower in IGL; however, because this analysis only included an assessment of lead guQTLs, we have not accounted for additional variants that may make additional genetic contributions. We previously showed that secondary guQTLs in IGH were able to increase the variance in gene usage explained by *cis* genetic factors[12]. As cohorts increase in size, we expect it will be possible to characterize additional guQTLs in IGK and IGL.

An assessment of guQTL positions within each locus provided initial evidence of the mechanisms by which they may exert their effects on the repertoire. Early studies in IGK were the first to show that IG polymorphism can directly impact the usage of particular genes in the repertoire. These specifically linked variation within the RSS of *IGKV2D-29* and its usage frequency[28,43,44], demonstrating that RSS polymorphisms have the potential to influence the binding of RAG1/2 and the selection of genes by V(D)J recombination. Here, we found additional direct evidence for effects of RSS variants on several IGK and IGL genes. However, consistent with previous observations in IGH[12], the majority of light chain guQTLs were intergenic SNVs. The factors underlying the effects of non-coding guQTLs require further study, but are expected to influence various regulatory mechanisms (e.g., enhancer and promoter function; formation of topologically associating domains) in the IG loci during V(D)J recombination[45–49]. It is notable that in IGL we observed overlap between a subset of intergenic guQTLs and known TFBSs (Supplemental Fig. 12); this included enrichments in binding sites for CTCF, a transcription factor known to be critical to the chromatin landscape within IG loci[50]. We also found

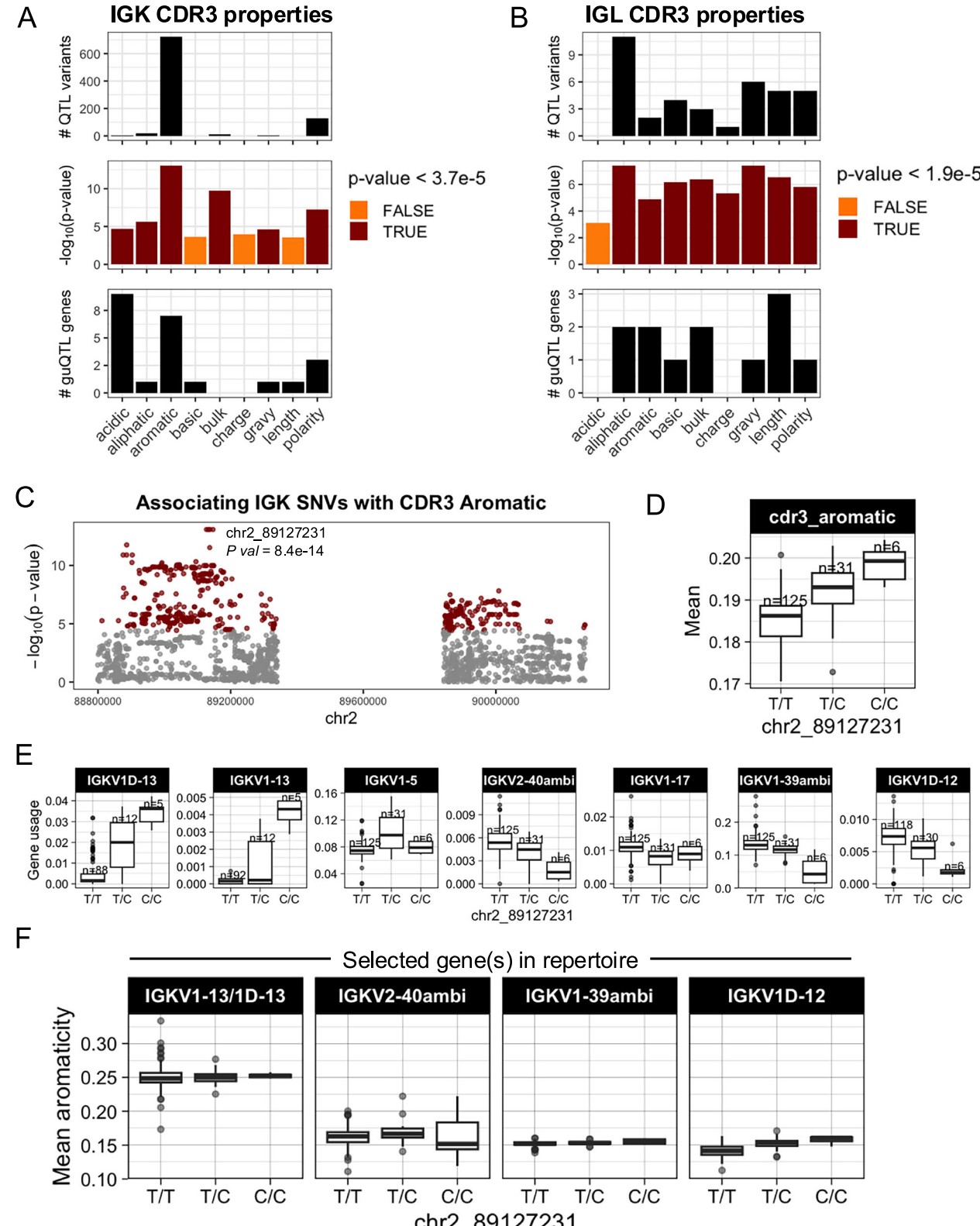

that SVs were the lead guQTLs for three genes (*IGKV1D-8*, *IGKV1-NL1*, and *IGLV5-39*). In all cases these SVs altered the number of diploid copies (range = 0–2) and thus the chance they could be selected by V(D)J recombination. We identified multiple less common SVs (MAF < 5%) that will require study in larger sample sizes to more fully assess their contribution to gene usage. Overall, however, we noted that the number of genes impacted by SVs in each of the light chain loci were comparatively fewer than reported in IGH[12], which is likely a

reflection of the fact that the IGH locus overall has a greater number of SVs.

We also observed examples in which guQTLs were localized to intronic and V gene coding sequences, the latter of which included examples resulting in the introduction of premature stop codons and amino acid changes. These examples indicate that, rather than effects on V(D)J recombination, some guQTLs potentially influence the light chain repertoire composition by impacting transcription and

**Fig. 6 | IGK and IGL variants impact CDR3 physicochemical properties in the naïve Ab repertoire. (A, B)** For each CDR3 physicochemical property (x-axis), mean values were computed for each individual and tested for association (linear regression) with all common variants in IGK (**A**) and IGL (**B**). Barplots show (i) the number of QTL variants (Bonferroni-corrected) for each property, (ii) the −log10(P value) for lead variants, and (iii) the number of guQTL genes identified for the lead CDR3 property QTL variant. Summary statistics are provided in Supplementary Data 12. **C** Manhattan plot shows the −log10(P value) for all SNVs in the IGK locus tested for association with CDR3 aromaticity, with QTLs colored dark red and the lead QTL labeled. **D** Boxplot of the mean IGK CDR3 aromaticity with individuals separated by genotype at the lead QTL. **E** Boxplots of usages for seven IGK genes that are guQTLs at the lead CDR3 aromaticity variant (linear regression; *P* value < 3.7e−5). **F** BCR sequences that used the indicated V genes were selected from the Ab repertoire, then mean CDR3 aromaticity of each repertoire subset was computed and plotted with individuals separated by genotype at the lead CDR3 aromaticity QTL. Boxplots display the median, 25th percentile, 75th percentile, and whiskers that extend up to 1.5 times the inter-quartile range (IQR) from the respective percentiles. Data points outside the whiskers are also plotted.

translation, with implications for light chain selection during B cell development. For example, we would expect that B cells expressing non-functional alleles would undergo receptor editing and/or negative selection in the bone marrow, leading to their absence in the periphery[51–53]. Likewise, it is plausible that some light chain coding alleles within an individual may serve as less optimal partners for rearranged heavy chains[54], leading to a decrease in their frequency within the mature naïve repertoire. This would be somewhat analogous to shifts in light chain gene distributions noted in the memory repertoire, which have been attributed to light chain coherence[55]. However, fully delineating the roles of genetics in heavy and light pairing and in the context of different antigen-driven effects, will require careful investigation of guQTLs across developmental time points, and will need to consider combined effects of polymorphisms across the three loci.

To date, differences in the genetic architecture of the IGK and IGL loci have been underexplored. Previous comparisons of a small number of IGK and IGL haplotypes indicated that SNV densities were lower in IGK compared to IGL[31]. Our data confirmed this pattern at the population level, revealing that the number of common SNVs was almost 2-fold higher in IGL. Additionally, we found that IGL was also characterized by less extensive LD. These stark differences in genetic architecture were reflected in the interconnectedness of gene usage profiles in the repertoire, specifically that a greater proportion of genes in the IGK repertoire shared overlapping guQTLs. Given these observations, we could expect the regulatory landscapes to also be different. To date, our knowledge of the regulation of V(D)J recombination in the light chain loci come from studies in mice; however, it is unlikely that we can extrapolate much detail from these studies, as these loci show little structural resemblance to those in humans[56]. Given the ordered engagement of IGK and IGL genes in the formation of functional BCRs during B cell development[51,56], it is plausible that the differences in genetic structure have been shaped by their differing functional roles. For example, it has been suggested previously that natural selection has favored the occurrence of inverted genes within the IGK locus as a means to facilitate repeated rounds of V(D)J recombination on a given haplotype during receptor editing[56]. Similarly, more recent work looking at IG locus genetic features across a range of vertebrate species suggest co-evolution of the light chain loci[57].

Throughout the human genome, more LD blocks per megabase of DNA implies higher recombination rates and haplotype diversity, and vice-versa[58–60]. The relatively large stretches of LD and lower density of common SNVs in IGK may reflect limited meiotic recombination. This should result in extended blocks of linked coding variants with more tightly orchestrated regulation of V(D)J recombination and gene usage profiles. By contrast, smaller LD blocks and a higher density of common SNVs in IGL are consistent with higher rates of meiotic recombination and haplotype diversity, which can provide opportunities for a wide range of potentially advantageous variants to persist within populations. Furthermore, it could be speculated that the risk of potentially deleterious variants in IGL is not a major expense to the overall probability of functional BCR generation as IGK is the initial source of expressed light chains. Instead, greater IGL haplotype

diversity could both serve as a source of diversity to broaden antigen recognition in the context of infection and/or provide a greater breadth of light chains for rescuing autoreactive BCRs. Early studies from a limited number of individuals showed that autoreactive antibodies were more likely to include an IGK chain, but could be rendered non-autoreactive by swapping in light chains encoded by alternative genes[61]. Interestingly, among the genes capable of reducing auto-reactivity (termed "editors") in select cases were *IGKV3-15, IGLV2-14* and *IGLV1-40*[61]; in our cohort, these genes were used at relatively high frequencies and were not associated with guQTLs (Supplementary Fig. 9). It is possible that selective pressures across populations have favored retention and higher usage of these genes, as they may reduce the likelihood of autoreactivity.

These data highlight the impacts of germline variation on gene usage variation in the repertoire, and their underlying mechanisms. We argue that the influence of genetics should be considered when seeking to understand how inter-individual differences in repertoire composition directly contribute to antigen-driven responses[19–22,24,62–67]. Given coding differences between genes, usage variation by default alters the landscape of available germline encoded residues among expressed BCRs, which can also bias SHM patterns[68]. Here, we demonstrate that guQTLs are also directly linked to amino acid differences between alleles of individual genes; thus, while some coding variants may be present within the genome of an individual, their frequency within the repertoire is dependent on guQTL genotype. Taking this a step further, we showed that gene usage also correlated with variation in CDR3 properties, driven by direct contributions of 3′ V and 5′ J gene germline-encoded bases to junction amino acids. This is consistent with observations of CDR3 comparisons between mono-zygotic twins and unrelated individuals[17]. The link between guQTLs and amino acid features among expressed light chain transcripts elevates the likelihood that guQTLs significantly impact the antigen-binding landscape of expressed Abs.

Together, these findings advance our basic understanding of repertoire development, illuminating regions of IGK and IGL that not only regulate gene usage but establish biases in the amino acid diversity observed among expressed Abs. In combination with our previous work in the IGH locus, our data lay a foundation for integrating genetic contributions from all three IG loci to establish more complete and precise models of sequence diversity in the expressed repertoire. This will be critical for refining our understanding of Ab repertoire dynamics in health and disease.

## Methods

### Sample information
PBMCs (*n* = 177) were procured from STEMCELL Technologies (Vancouver, Canada). Sample-level demographic information, including age, biological sex, and ancestry informative marker (AIM)-determined ancestry are reported in Supplementary Data 1.

### Single-molecule real-time (SMRT) long-read library preparation and sequencing
DNA was extracted from ~3–5 million PBMCs per donor using the DNA/RNA co-extraction AllPrep kit (Qiagen, Germantown, MD, USA), and

genomic DNA was processed using our published "IG-capture" targeted long-read sequencing protocol[12–15]. Briefly, high molecular weight DNA (~2.5 µg) was sheared to ~15 Kbp using g-tubes (Covaris, Woburn, MA, USA) and size-selected using Pippin systems (Sage Science, Beverly, MA, USA) using the "high pass" protocol to select fragments greater than 5 Kbp. Size-selected DNA was ligated to universal barcoded adapters and amplified, and small fragments and excess reagents were removed using 0.7X KAPA Pure beads (Roche, Indianapolis, IN, USA). Individual samples were pooled in groups of six prior to IGK and IGL enrichment using custom Roche HyperCap DNA probes described previously[14,15]. Targeted fragments were amplified after capture to increase total mass for sequencing library construction.

Enriched IGK and IGL libraries were prepared for sequencing using the SMRTBell Express Template Prep Kit 2.0 (Pacific Biosciences, Menlo Park, CA, USA) and SMRTBell Enzyme Cleanup Kit 1.0 (Pacific Biosciences), according to the manufacturer's protocol. Resulting SMRTbell libraries were multiplexed in pools of 12 and sequenced using one SMRT cell 8 M on the Sequel IIe system ($n = 134$) (Pacific Biosciences) using 2.0 chemistry and 30-hour movies. For Revio sequencing ($n = 43$), SMRTbell libraries were pooled in 36-plexes and sequenced using one SMRT cell 25 M on the Revio system (Pacific Biosciences) using Revio Polymerase Kit v1.0 (PacBio; 102-739-100) and 30 hour movies. High Fidelity ("HiFi") intramolecular circular consensus reads with accuracies >99.9% (Q20) were generated on instrument and used for all downstream analyses.

### IGK and IGL AIRR-seq
AIRR-seq libraries were prepared and sequenced for 164 individuals (IGK) and 168 individuals (IGL). RNA was extracted from ~3-5 million PBMCs per donor using the AllPrep DNA/RNA Kit (Qiagen). AIRR-seq libraries were generated using a 5' Rapid Amplification of cDNA Ends (RACE) approach. For IGK and IGL 5'RACE AIRR-seq, libraries were produced using the SMARTer Human BCR Profiling Kit (Takara Bio, San Jose, CA, USA), according to the manufacturer's instructions. Quality and quantity of individually indexed IGK libraries were determined using the 2100 Bioanalyzer High Sensitivity DNA Assay Kit (Agilent, Santa Clara, CA, USA) and IGL libraries with the Qubit 3.0 Fluorometer dsDNA High Sensitivity Assay Kit (Life Technologies, Carlsbad, CA, USA). Libraries were pooled at 10 nM and sequenced on the Illumina NextSeq system using 300 bp paired-end reads with the 600-cycle NextSeq P1 Reagent Kit ($n = 160$) or with the 600-cycle MiSeq Reagent Kit v3 ($n = 11$) (Illumina, San Diego, CA, USA).

### Construction of a custom linear reference assembly
We used our previously described custom linear reference[15], which includes modifications to the IGH[12,13] and IGK[15] loci to include sequences not present in the GRCh38 assembly. To include IGL sequences not present in GRCh38, chromosome 22 was removed and replaced with the T2T (CHM13v2.0) chromosome 22, including the *IGLV5-39* structural variant (SV, insertion) sequence. In addition, 46,423 bp of sequence (chr22:23315600-23362023) is from a Human Pangenome Reference Consortium (HPRC) haplotype from sample HG00621, which includes a 16,093 bp insertion relative to the CHM13v2.0 reference, reflecting 3 additional copies of the IGLJ-C3 cassette relative to the CHM13v2.0 reference. This reference is publicly available at https://github.com/Watson-IG/immune_receptor_genomics/tree/main.

### Phased assembly of IGK and IGL
Phased assemblies were generated as described previously[15]. HiFi reads were used to generate haplotype-phased de novo (i.e. reference-agnostic) assemblies using hifiasm[69] (v0.18.2-r467) with default parameters. For each sample, hifiasm contigs were concatenated into a FASTA file, then redundant contigs were filtered out using the seqkit toolkit[70] command 'seqkit rmdup --by-seq <hifiasm_contigs.fasta >'

(seqkit v2.4.0). Hifiasm contigs were mapped to the custom reference assembly using minimap2 (v2.26) with the '-x asm20' option. HiFi reads were also processed using IGenotyper[13]; the programs "phase" and "assemble" were run with default parameters to generate phased contigs and HiFi read alignments to the custom reference.

For each sample, aligned HiFi reads as well as aligned hifiasm-generated contigs and IGenotyper-generated contigs were viewed in the Integrative Genomics Viewer (IGV) application[71] for manual selection of phased contigs. Contigs were evaluated for read support from mapped HiFi reads, and contigs harboring one or more SNVs that lacked read support were not selected during manual curation. Where a hifiasm and an IGenotyper contig were identical throughout a phased block, the hifiasm contig was selected, as described previously[15]. Curated, phased assemblies were aligned to the custom reference using minimap2[72] with the '-x asm20' option.

To assess accuracy of manually curated assemblies, personalized references were first generated by N-masking the IGK (chr2:88837160-90280100) and IGL (chr22:22378775-23423320) loci of the custom reference and, for each sample, appending the reference FASTA with IGK and IGL curated contigs. All HiFi reads from each individual were aligned to the corresponding personalized reference using minimap2 with the '-x map-hifi' preset; coverage and read length metrics were extracted from these alignments. Positions in assemblies with > 25% of aligned HiFi reads mismatching the assembly were identified by parsing the output of samtools (v1.17) 'mpileup'; assembly accuracy was determined using the formula [total (diploid) bases without a mismatch/total (diploid) assembly length (bp)] * 100 = % accuracy (Supplementary Data 1). These scripts, including our custom reference assembly, are available at https://github.com/Watson-IG/wasp.

### Identification of IGK and IGL gene alleles
Sequences (alleles) corresponding to V, J, and C exons were obtained from assembly BAM files using code available at https://github.com/Watson-IG/wasp; these scripts include metrics of read support from HiFi reads. Briefly, for each sample, curated IGK and IGL assembly contigs were appended to our custom reference with the IGK and IGL loci N-masked to generate a personalized reference FASTA, as described above. All HiFi reads from IG-capture for a given individual were mapped to the personalized reference using minimap2 with the '-ax map-hifi' preset, then the resulting BAM file was input to samtools 'mpileup' with iteration over allele coordinates in the personalized reference. The outputs of this script are in Supplementary Data 3 (IGK) and Supplementary Data 4 (IGL), and include a column for total number of HiFi reads spanning the allele ('Fully_Spanning_Reads') and a column for the number of HiFi reads spanning the allele with 100% sequence identity ('Fully_Spanning_Reads_100%_Match'). A complete description of HiFi read support metrics for alleles is available at https://vdjbase.org/.

### Genetic ancestry
IGenotyper[13] was used to call SNVs at ancestry-informative markers (AIMs) by aligning, phasing, and locally assembling reads at AIM regions, then directly identifying SNVs from the assembled sequences. Genetic ancestry was determined using these AIMs and the STRUCTURE program[73]. SNV VCFs were processed to extract AIM-specific data from IG-Capture libraries using vcftools (v0.1.16; code available at https://github.com/oscarlr/IGenotyper/tree/master/IGenotyper/ancestry). Coverage of AIMs was assessed using BAM files and the pysam (v0.21.0) library, ensuring a minimum read depth threshold for inclusion. Genotypes were converted into haplotypes by separating phased alleles, and samples were coded alongside reference populations from the 1000 Genomes Project. STRUCTURE (v2.3.4) was run with $K = 5$, representing five global ancestry groups (European, African, East Asian, South Asian, and American), using default admixture and allele frequency models. For each sample, the two highest ancestry

proportions were identified. If the difference between these two proportions was less than or equal to 5%, the sample was classified as "Mixed." Otherwise, the sample was assigned to the ancestry category with the highest proportion.

## Processing AIRR-seq data and calculating gene usage

Paired-end sequences ("R1" and "R2") were processed using the pRE-STO/Change-O toolkit[74,75]. All R1 and R2 reads were trimmed to $Q = 20$ using the function "FilterSeq.py trimqual". Constant region (IGKC and IGLC) primers were identified with an error rate of 0.3 and corresponding chains were recorded in the fastq headers using "MaskPrimers align." The 12 base UMI, located directly after the constant region primer, was extracted using "MaskPrimers extract." Annotations between mate pairs, including UMI barcodes and constant region calls, were synchronized using PairSeq.py to sort reads into mate pairs and remove unpaired reads.

UMI groups sharing the same barcode were processed to generate consensus sequences using the BuildConsensus.py function. The following criteria were applied: a minimum UMI group size of one, a maximum mismatch error rate of 10%, and at least 60% agreement on the constant region call within the group. Reads with lower-quality consensus sequences ($Q < 30$) were masked using FilterSeq.py maskqual. Duplicate counts ("Dupcounts") were recorded, and duplicate sequences were collapsed using CollapseSeq.py to retain one representative sequence per cell, with the total number of sequences contributing to each consensus recorded as "Conscount." Collapsed consensus sequences supported by fewer than two contributing reads (Conscount < 2) were discarded using SplitSeq.py. Samples containing fewer than 100 unique sequences were excluded from downstream analysis; after filtering, 164 individuals were included for IGK analyses and 168 individuals were included for IGL analyses (171 individuals total; Supplementary Data 1). After processing, the repertoires contained a mean of 33,167 unique BCR sequences (IGK) and 13,310 (IGL) (Supplementary Data 1, Supplementary Fig. 3).

Germline allele designations were assigned to sequences using a personalized allele database during the IgBLAST step. For each individual, IGK and IGL germline allele databases were generated from the set of alleles identified in genomic assemblies derived from long-read sequencing. Separate BLAST databases were created for V and J segments using makeblastdb (v2.5.0). The resulting databases were used as input for the igblastn function in the Change-O toolkit (v1.3.01)[75], and IgBLAST output files ("Change-O" tables) of unique BCRs were generated for IGK and IGL separately. This process permitted theoretical disambiguation of IGK gene paralogs for individuals wherein the sequence of each allele of a proximal paralog was distinct from the sequence of each allele of the distal paralog (results in Supplementary Data 5). In the case of *IGKV1-13* and *IGKV1D-13*, a subset of the cohort (n = 104) met this disambiguation criteria and was carried forward to identify guQTLs for these genes.

Due to sequence identity between IGKV paralog allele sequences, genes collapsed into a single ambiguous ("ambi") entity in Change-O tables included:
1) *IGKV1-37* and *IGKV1D-37* replaced with *IGKV1-37ambi*
2) *IGKV1-39* and *IGKV1D-39* with *IGKV1-39ambi*
3) *IGKV2-40* and *IGKV2D-40* with *IGKV2-40ambi*
4) *IGKV1-33* and *IGKV1D-33* with *IGKV1-33ambi*
5) *IGKV2-28* and *IGKV2D-28* with *IGKV2-28ambi*

In addition, IGLJ gene calls that corresponded to a IGLJ2 or IGLJ3 cassette (*IGLJ2, IGLJ3-1, IGLJ3-2, IGLJ3-3, IGLJ3-4*) were collapsed to "*IGLJ2-3ambi*".

Change-O tables of unique BCRs were analyzed using the alakazam (1.3.0) package[75]. To enrich for antigen-naïve BCRs, only unmutated light chain sequences with 100% identity to the assigned germline V and J alleles were included in downstream analyses. For analysis of mutated sequences (i.e. the fraction of sequences enriched with those containing SHM), sequences with less than 100% identity to either the germline V or J allele were included. Gene usage was quantified for IGK and IGL light chains using the countGenes function with the parameters gene = "v_call" and "j_call", groups = "sample_id", mode = "gene", and genes with a sequence count (seq_count) of at least 10 in at least one sample were retained. A m × n usage matrix C was created, where m are the genes and n are the samples. Each value in C represented usage frequency among all unique (unmutated or mutated) BCR sequences for a given gene in a given sample.

CDR3 physicochemical properties were computed using the aminoAcidProperties function[75] with options seq = "junction", trim = TRUE, label = "cdr3"; resulting values were then averaged across all sequences within each sample to obtain sample-level means.

## Selecting common variants for gene usage QTL analysis

SNVs were genotyped from curated assemblies as described previously[15]. Variants were called from assemblies using 'bcftools mpileup' (bcftools v1.15.1) with options '-f -B -a QS', then 'bcftools call' with options '-m --ploidy 2'. A mutli-sample VCF file was generated using 'bcftools merge' with the '-m both' option. Multiallelic SNVs were split into biallelic records using the 'bcftools norm' command with options '-a -m-'. The VCF file was annotated for V-exon, introns, L-Part1, RSS sequences (heptamer, nonamer, spacer), were added using vcfanno[76] (v0.3.3) and BED files corresponding to our reference (available at https://github.com/Watson-IG/immune_receptor_genomics/). Biallelic SNVs with MAF ≥ 5% were selected using bcftools view with options "-m2 -M2 -v snps -i 'INFO/MAF > = 0.05'" and used for guQTL analysis.

All SVs in IGKV and IGLV gene regions were genotyped by manual inspection using IGV[71]. SVs with a MAF less than 0.05 were not included in the guQTL analysis.

## Gene usage QTL analysis

Genotypes at common SNVs and SVs were tested for association with usage using linear regression to determine significance and additional metrics (e.g., beta coefficients and $R^2$ values) using R (v4.3.1). To adjust for multiple comparisons, a Bonferroni correction was applied on a per-gene basis. Pairwise $r^2$ (LD) values were computed using vcftools '--geno-r2'[77]. Variants in complete linkage disequilibrium (LD $r^2 = 1$) were considered as a single variant during correction, representing only one association test.

## Haplotype block analysis

LD blocks were computed and visualized using "LDBlockShow"[39] using the multi-sample VCF of common SNVs (MAF ≥ 5%) as input, with options '-SeleVar 2 -BlockType 1' to use normalized linkage disequilibrium coefficients (D') as described by described by Gabriel et al. (2002)[40] to determine haplotype blocks. The program was run using '-Region chr2:88837160-90280100' for IGK and '-Region chr22:22378775-23423320' for IGL. LD block boundaries, sizes, and SNVs within blocks are included in Supplementary Data 9. Genes overlapping LD blocks were identified using bedtools[78] 'intersect' (bedtools v2.30.0).

## Network analysis

Variants significantly associated with IGK and IGL gene usage (after Bonferroni correction) were compiled to create a set of guQTL variants for each gene. Pairwise comparisons between genes were performed to calculate Jaccard similarity indices, defined as the ratio of shared variants to the total unique variants across the two genes being compared. Only gene pairs with nonzero similarity (i.e., at least one shared variant) were included in the analysis. Separate IGK and IGL network graphs were constructed and visualized using

igraph[79] and ggraph[80] (IGL),. In these graphs, nodes represented individual genes and were positioned using the Fruchterman-Reingold force-directed algorithm, edges connected pairs of genes that shared guQTL variants, and edges were labeled to reflect the number of shared guQTL variants.

## Regulatory analysis

ENCODE transcription factor binding site data were obtained from the UCSC Genome Browser under the group "Regulation," track "TF Clusters," and table "encRegTfbsClustered." SNVs associated with gene usage were overlapped with this track and enrichment over all SNVs overlapping each track was calculated using a one-sided Fisher Exact Test (Supplementary Data 8).

## Reporting summary

Further information on research design is available in the Nature Portfolio Reporting Summary linked to this article.

## Data availability

Long-read sequencing data and AIRR-seq datasets generated in this study have been deposited in the BioProject repository https://www.ncbi.nlm.nih.gov/bioproject/?term=PRJNA1274485. Previously published AIRR-seq datasets are available in the BioProject repository https://www.ncbi.nlm.nih.gov/bioproject/?term=PRJNA555323. Metadata and sequencing summary statistics for this study are provided in Supplementary Data 1.

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

## Acknowledgements

The authors are grateful for constructive feedback provided by two anonymous reviewers as well as Steven Ellis and Ronald Gregg at the University of Louisville School of Medicine. This work was supported by grant R24AI138963 to C.T.W. and M.L.S. from the National Institute of Allergy and Infectious Disease. SMRT sequencing on the Revio system was supported by grant S10OD034432-01 to M.L.S. from the National Institutes of Health.

## Author contributions

E.E., O.L.R., M.L.S., and C.T.W. contributed to the conception of the work. M.L.S. and C.T.W. directed wet lab/bench and sequencing experiments. E.E., D.R.S., K.S., S.Schultze. prepared sequencing libraries. E.E., U.J., O.L.R., W.L., W.S.G., A.P., G.Y. S.Saha., and Z.V. developed the computational pipeline to analyze the sequencing data. E.E., G.Y., C.T.W., M.L.S contributed to and supervised the analysis and data interpretation. C.T.W. and M.L.S. acquired funding. E.E. and C.T.W. drafted the manuscript and all authors contributed to revising the manuscript.

## Competing interests

C.T.W., M.L.S., W.L. and G.Y. are shareholders of Clareo Biosciences, Inc.; C.T.W., M.L.S. and W.L. serve on the executive board and G.Y. is an advisor. The remaining authors declare no competing interests.
