## [Peer Review File · Nature Communications]

Germline polymorphisms in the immunoglobulin kappa and lambda loci underpinning antibody light chain repertoire variability

Corresponding Author: Dr Corey Watson

Version 0:

Reviewer comments:

Reviewer #1

(Remarks to the Author)

Engelbrecht et al. present an in-depth study of genetic variation in the IGK and IGL loci and its impact on antibody repertoires. This study complements their previous work on IGH and contributes to a more complete understanding of immunoglobulin gene regulation. The use of long-read sequencing, paired with AIRR-seq data from over 170 genetically diverse individuals, enables a uniquely detailed analysis of IGK and IGL variation and its functional consequences. The authors convincingly demonstrate cis-regulatory effects of germline variants on gene usage, including associations with CDR3 features relevant to antibody specificity and structure. The identification of more than 300 novel IG alleles represents a major contribution that will enrich current germline reference databases. The integration of genotype and repertoire data is thoughtfully executed, with clear evidence of variant effects in both naive and antigen-experienced B cell compartments. Additionally, the analyses of linkage disequilibrium and gene co-regulation offer valuable insights into structural and evolutionary distinctions between the IGK and IGL loci. I have only a few minor comments.

1. The use of “preclude” in the second sentence of the Abstract may be misleading, as it implies prevention of antibody function. Consider replacing it to more accurately reflect the findings.
2. Please introduce the abbreviation SNV (single nucleotide variant) in full at first mention for clarity.
3. Consider minimizing dense numerical detail in the main text (e.g., variant counts, percentages, LD block sizes) to improve readability and maintain narrative flow.
4. Discussion, line 411: The mention of D genes in the context of light chains seems out of place and may be a typographical oversight.
5. The paper clearly demonstrates genomic and regulatory differences between IGK and IGL (e.g., LD structure, guQTL density, co-regulation). It would be helpful to include a brief discussion of the potential functional implications of these differences for antibody diversity and immune responses. The authors refer to “differing functional roles” of kappa and lambda light chains (line 473); clarification or supporting context for this statement would strengthen the discussion.

Reviewer #2

(Remarks to the Author)

In this paper, the investigators used long-read sequencing of the antibody Igk and Igl light chain repertoires with Ig repertoire transcriptome sequencing. This allowed a comparison of light chain repertoire polymorphism, including features such as genetic variation, plus gene allele and nucleotide or RSS differences, to the expressed repertoire. Using a similar approach, they had previously reported that IgH polymorphism strongly impacted IgH expression.

Comments:

The significance of this study is understated, and essential perspectives regarding the role of heavy versus light chains in BCR repertoire development are overlooked. In thinking of this question from an immunologist’s perspective, to me, predictably, genetics should dictate IgH repertoire usage. But this is not the case for light chains that that should be used with greater selective pressures. The IgH rearranges first, and limitations due to the removal of surrounding D genes and properly directed RSS limit IgH rearrangement to essentially a single try, unless out of frame, then a 2nd allele if needed. Thus, the IgH repertoire is fixed. It is then the flexibility of light chain rearrangement that fits to and shapes the final antibody repertoire. Thus, the light chain repertoire is expected to be used in a fashion much more driven by the immovable IgH

repertoire and so may not be as reflective of the genetics. This is facilitated by the lack of a D chain, which allows repeated rearrangements of a single allele in a leap-frog fashion, and by having four alleles of light chains that progress through kappa first and then lambda. Thus, it is well appreciated that there is a necessity for repeated light chain rearrangements to “fix” antibodies with poor HC/LC pairings and for receptor editing, to alter autoimmune BCRs. Specific things that should be considered in interpreting this data:

1. In fact, most mature B cells have IgH chains paired with light chains, resulting from multiple rearrangements that have occurred (i.e., Melchers and colleagues, Immunity, 1999). As this occurs progressively through the kappa alleles with lambda as the final option, could this drive differential usage of lambda chains?
2. Receptor editing: classic work from labs like Martin Weigert's has shown that progressive usage of particular light chains is used for repairing auto-reactive and polyreactive BCRs. These include “Master editors” from both the kappa and lambda alleles that tend to balance charge, for example, and so should occur at frequencies disproportionate to what genetics predicts. Is this the case?
3. Half of the human kappa repertoire is an inverted copy of the other, such that downstream rearrangements “flip” the entire first half of the IgK chains, leaving them now distinctly positioned. How does this impact usage?

Version 1:

Reviewer comments:

Reviewer #1

(Remarks to the Author)

The authors have addressed the comments and revised the manuscript accordingly. The responses are clear and the revisions improve the clarity and presentation of the work.

Reviewer #2

(Remarks to the Author)

All concerns were adequately addressed.

Manuscript NCOMMS-25-50865-T: "Germline polymorphisms in the immunoglobulin kappa and lambda loci explain variation in the expressed light chain antibody repertoire", Engelbrecht et al.

Response to reviewer comments

We were delighted to read such supportive reviews, and are very grateful for the thoughtful feedback provided by both reviewers. Please find detailed responses to each comment below.

REVIEWER COMMENTS

Reviewer #1 (Remarks to the Author):

Engelbrecht et al. present an in-depth study of genetic variation in the IGK and IGL loci and its impact on antibody repertoires. This study complements their previous work on IGH and contributes to a more complete understanding of immunoglobulin gene regulation. The use of long-read sequencing, paired with AIRR-seq data from over 170 genetically diverse individuals, enables a uniquely detailed analysis of IGK and IGL variation and its functional consequences. The authors convincingly demonstrate cis-regulatory effects of germline variants on gene usage, including associations with CDR3 features relevant to antibody specificity and structure. The identification of more than 300 novel IG alleles represents a major contribution that will enrich current germline reference databases. The integration of genotype and repertoire data is thoughtfully executed, with clear evidence of variant effects in both naive and antigen-experienced B cell compartments. Additionally, the analyses of linkage disequilibrium and gene co-regulation offer valuable insights into structural and evolutionary distinctions between the IGK and IGL loci. I have only a few minor comments.

1. The use of "preclude" in the second sentence of the Abstract may be misleading, as it implies prevention of antibody function. Consider replacing it to more accurately reflect the findings.

We agree that "preclude" in this sentence is an interpretation that over-extends our findings, and have therefore changed the language to "influence".

2. Please introduce the abbreviation SNV (single nucleotide variant) in full at first mention for clarity.

We have addressed this change in our revised manuscript.

3. Consider minimizing dense numerical detail in the main text (e.g., variant counts, percentages, LD block sizes) to improve readability and maintain narrative flow.

We thank the reviewer for this feedback. We have attempted to shorten several paragraphs within our results to address this comment.

4. Discussion, line 411: The mention of D genes in the context of light chains seems out of place and may be a typographical oversight.

Thank you for pointing out this oversight, we have removed mention of "D" genes at this point in the Discussion.

5. The paper clearly demonstrates genomic and regulatory differences between IGK and IGL (e.g., LD structure, guQTL density, co-regulation). It would be helpful to include a brief discussion of the potential functional implications of these differences for antibody diversity and immune responses. The authors refer to "differing functional roles" of kappa and lambda light chains (line 473); clarification or supporting context for this statement would strengthen the discussion.

Thank you for this suggestion. We have expanded on this line of thought in sentences that follow with examples of how LD structure, SNV density, and co-regulation can impact IGK versus IGL repertoires. We also offer potential explanations for the observed differences in LD structure and SNV densities.

Reviewer #2 (Remarks to the Author):

In this paper, the investigators used long-read sequencing of the antibody Igk and Igl light chain repertoires with Ig repertoire transcriptome sequencing. This allowed a comparison of light chain repertoire polymorphism, including features such as genetic variation, plus gene allele and nucleotide or RSS differences, to the expressed repertoire. Using a similar approach, they had previously reported that IgH polymorphism strongly impacted IgH expression.

Comments:

The significance of this study is understated, and essential perspectives regarding the role of heavy versus light chains in BCR repertoire development are overlooked. In thinking of this question from an immunologist's perspective, to me, predictably, genetics should dictate IgH repertoire usage. But this is not the case for light chains that should be used with greater selective pressures. The IgH rearranges first, and limitations due to the removal of surrounding D genes and properly directed RSS limit IgH rearrangement to essentially a single try, unless out of frame, then a 2nd allele if needed. Thus, the IgH repertoire is fixed. It is then the flexibility of light chain rearrangement that fits to and shapes the final antibody repertoire. Thus, the light chain repertoire is expected to be used in a fashion much more driven by the immovable IgH repertoire and so may not be as reflective of the genetics. This is facilitated by the lack of a D chain, which allows repeated rearrangements of a single allele in a leap-frog fashion, and by having four alleles of light chains that progress through kappa first and then lambda. Thus, it is well appreciated that there is a necessity for repeated light chain rearrangements to "fix" antibodies with poor HC/LC pairings and for receptor editing, to alter autoimmune BCRs. Specific things that should be considered in interpreting this data:

1. In fact, most mature B cells have IgH chains paired with light chains, resulting from multiple rearrangements that have occurred (i.e., Melchers and colleagues, Immunity, 1999). As this occurs progressively through the kappa alleles with lambda as the final option, could this drive differential usage of lambda chains?

We appreciate this question. It sounds like the reviewer is inquiring: "does the process of kappa rearrangement influence usage of lambda genes?" We apologize if this is a misinterpretation of the question.

We agree with the commonly accepted statement that kappa alleles are rearranged first, followed by lambda, and mention this with citations in our revised Discussion. We also speculate on how IGK genetic variation impacts the repertoire as compared with IGL genetic variation in our revised Discussion.

Given that our AIRR dataset is bulk sequencing of BCR mRNAs from the periphery, we cannot directly compute IGK genomic rearrangements in individual B cells. It is certainly possible that IGK germline sequence impacts the number of IGK rearrangements that occur within a given B cell as well as the probability of switching to lambda. In unpublished data, we see evidence that germline variants in IGK can associate with usage of IGL genes (i.e. *trans* associations). This result in general might indicate that the presence in the genome and preferred usage of particular IGK genes could modulate the frequency at which receptor editing drives toward the use of lambda – in such a scenario, we could expect to see that IGK variants would have secondary effects on IGL genes. This may also be consistent with the data presented in our current manuscript that usage variation in some IGK genes is

largely explained by single genetic variants, while other genes, and especially in IGL have more modest effect sizes. Stated another way, the model we are building toward is one in which each of the three loci harbor genetic variants that modulate and shift V(D)J recombination frequencies of particular genes (the effects of which would be observed most clearly in early developmental B cell stages, pro- and pre- B cells, rather than in periphery, where our data are drawn from). Once these initial (primarily genetic) biases are established in the early repertoire, there is an interplay of both genetics and developmental selection that occurs that will result in additional shifts that we expect are also very likely to be predicted in part by genetics. Shifts in heavy and light chain repertoires would then be expected to be dependent on both locus-specific as well as intra-locus factors, dictated by processes like the ordered recombination of loci mentioned here, as well as, e.g., the need of the repertoire to “find” the most optimal pairings, drawing from the available chains put into the repertoire initially by IGH and IGK, followed by IGL. In future studies, we plan on investigating associations between germline sequence and rearrangements in single-cell repertoires, and across developmental timepoints, which will allow us to more directly address these questions, and refine this proposed model.

2. Receptor editing: classic work from labs like Martin Weigert’s has shown that progressive usage of particular light chains is used for repairing auto-reactive and polyreactive BCRs. These include “Master editors” from both the kappa and lambda alleles that tend to balance charge, for example, and so should occur at frequencies disproportionate to what genetics predicts. Is this the case?

Thank you for this question. Because this study is currently focused on BCRs in the periphery, we are technically seeing the result of repertoires that have likely been influenced uniquely and in combination by genetic effects on V(D)J recombination as well as pairing and selection, all of which we expect to make significant contributions during B cell development within the bone marrow. While not comprehensive in terms of power, we have recently released a preprint in which we have demonstrably shown that variants within IGH do in fact directly impact V(D)J recombination early in development (<https://www.biorxiv.org/content/10.1101/2025.05.19.654982v1>). Future studies will need to address similar questions for light chain guQTLs. However, what we find most remarkable about the data we’ve presented in our current manuscript is that, perhaps in contrast to earlier viewpoints/assumptions, impacts of genetics on the light chain repertoire do persist even into the periphery. Again, we expect that the mechanisms underlying these genetic effects are likely multi-factorial. Furthermore, regarding reference to “Master editors”, it is interesting that we see genes in the IGK and IGL repertoires that are not significantly associated with *cis* genetic variants, shown in Figure S9. For example, we did not identify *cis* guQTLs for IGKV3-15, IGLV2-14, or IGLV1-40, each of which are used at relatively high frequencies in their respective light chain repertoires. It is possible that these genes pair favorably (avoid autoreactivity) with a wide range of heavy chains, as was suggested in a study of “editors” in the context of autoreactivity in humans (Wardemann H, et. al, JEM 2004), which we cite in our revised Discussion. Perhaps any genetic effects on these genes would only be observed early in development, prior to selection-associated effects that would override the V(D)J recombination-related genetic signals, and result in these genes ending up at “unexpected” frequencies in the periphery, regardless of their “genetic predictions”. These questions could likely be addressed by profiling these genes through development.

Additionally, it is possible that negative evolutionary selection pressures at the species level have acted on these genes and their associated regulatory elements, limiting effects of *cis*-acting variants. And finally, although we did not observe *cis*-acting variants here, it is also possible that *trans*-acting

variants (e.g. variants in IGH) impact usage of these genes by a mechanism that involves biased selection of particular heavy chains, subsequently influencing light chains that can effectively pair. We are actively conducting studies of *trans*-acting genetic effects on the antibody repertoire – the complexity of these analyses and the observations we are making, however, place them beyond the scope of the current manuscript.

3. Half of the human kappa repertoire is an inverted copy of the other, such that downstream rearrangements “flip” the entire first half of the IgK chains, leaving them now distinctly positioned. How does this impact usage?

We agree that this is an interesting question, and have addressed it in Figure S14 titled “Analysis of IGKV gene paralog usages” and the associated Supplemental Text section titled “Genetic variants disrupt biases in differential usage of IGKV proximal and distal paralogs.” This section compares usages of proximal-distal paralog pairs. Proximal paralogs are generally used at a higher frequency, with the exceptions of IGKV2-29/2D-29 and IGKV1-13/1D-13. Mechanisms underlying these exceptions are discussed in Figure 2 (IGKV2-29), Figures S14 and S15, and associated text.